# Bacteriocin-Producing Strain *Lactiplantibacillus plantarum* LP17L/1 Isolated from Traditional Stored Ewe’s Milk Cheese and Its Beneficial Potential

**DOI:** 10.3390/foods11070959

**Published:** 2022-03-25

**Authors:** Andrea Lauková, Martin Tomáška, Maria Joao Fraqueza, Renáta Szabóová, Eva Bino, Jana Ščerbová, Monika Pogány Simonová, Emília Dvorožňáková

**Affiliations:** 1Centre of Biosciences of the Slovak Academy of Sciences, Institute of Animal Physiology, Šoltésovej 4–6, 040 01 Košice, Slovakia; renata.szaboova@uvlf.sk (R.S.); bino@saske.sk (E.B.); scerbova@saske.sk (J.Š.); simonova@saske.sk (M.P.S.); 2Dairy Research Institute, a.s., Dlhá 95, 010 01 Žilina, Slovakia; tomaska@vumza.sk; 3Faculty of Veterinary Medicine, University of Lisbon, Avenida da Universidade Tecnica, 1300-477 Lisbon, Portugal; mjoaofraqueza@fmv.ulisboa.pt; 4Parasitological Institute of the Slovak Academy of Sciences, Hlinkova 3, 040 01 Košice, Slovakia; dovoroz@saske.sk

**Keywords:** stored ewe’s cheese, source, beneficial bacteria, lactic acid bacteria

## Abstract

Stored ewe’s milk lump cheese is a local product that can be a source of autochthonous beneficial microbiota, especially lactic acid bacteria. The aim of this study was to show the antimicrobial potential of *Lactiplantibacillus plantarum* LP17L/1 isolated from stored ewe’s milk lump cheese. *Lpb. plantarum* LP17L/1 is a non-hemolytic, non-biofilm-forming strain, susceptible to antibiotics. It contains genes for 10 bacteriocins—plantaricins and exerted active bacteriocin with in vitro anti-staphylococcal and anti-listerial effect. It does not produce damaging enzymes, but it produces β-galactosidase. It also sufficiently survives in Balb/c mice without side effects which indicate its safety. Moreover, a reduction in coliforms in mice jejunum was noted. LP17L/1 is supposed to be a promising additive for Slovak local dairy products.

## 1. Introduction

Ewe’s milk and products made from ewe’s milk remain irreplaceable in human nutrition [1]. In Slovakia, ewe’s milk products are very popular because sheep breeding has a long tradition. Sheep are mostly grazed on mountainous pastures, so ewe’s milk can be supposed as a biofood. Ewe’s milk contains more protein, vitamins, and trace elements compared with cow milk [1]. Many traditional products made from ewe’s milk such as ”Slovenská parenica“, “Slovenská bryndza“, and “Slovenský oštiepok“ have been designed as protected geographical products (PGI), meaning with protected geographical indication since 2008. Ewe’s milk lump cheese received a traditional specialty guaranteed (TSG) label [2]. This indication designated by the European Union Commission was decided because of the special traditional manufacturing of ewe’s milk lump cheese. Moreover, stored ewe’s milk lump cheese belongs to these products. These products can be a source of autochthonous beneficial microbiota especially lactic acid bacteria, mostly from the phylum Firmicutes such as lactobacilli, lactococci, streptococci, and/or pediococci [3,4]. Besides this, stored ewe’s milk lump cheese is a base component for bryndza production [1]. Ewe’s milk lump cheese is processed as formerly described by Lauková et al. [5] but cheese for stored cheese processing has to be pressed for longer, then it is minced and salted (to have a salt concentration of around 4–6%) in special wood barrels which are lined with wood veneer. Barrels are filled, closed, and placed in a cellar for two months for ripening at temperatures from 2–6 °C.

The species *Lactobacillus plantarum* (now *Lactiplantibacillus plantarum*) [6] has been reported several times as a probiotic additive [7,8] which can also produce bacteriocin [9]. Bacteriocins are low molecular, thermo-stable, antimicrobial, and ribosomal active peptides which are synthesized by many species of bacteria including lactic acid bacteria (LAB). They show antimicrobial activities against food pathogens [10,11]. Many lactobacilli with probiotic characteristics are described as functional additives for dairy products [12,13]. Molecular studies reveal that bacteriocin determinants are mostly grouped in operons-regulons [11]. Structural genes for bacteriocin production can be found on plasmids, transposons, or mobile genetic elements inside the bacterial chromosome [14]. Bacteriocins produced by different food-originated LAB of many species have been reported up to now [11]. Their benefit is not only as an additive to beneficially influence the product itself but they can be beneficial for consumers via products such as, e.g., cholesterol-reducing functional bacteria [12]. Bacteriocin-producing bacteria are used as the main starter or adjunct culture for making cheese or fermented dairy products to prevent spoilage bacteria [11]. Strains of the species *Lactococcus lactis* and/or *Lactobacillus casei* are mostly used in the dairy industry [11]. This study focused on the antimicrobial potential of *Lpb. plantarum* strain LP17L/1 originating from stored ewe’s milk lump cheese for its further application possibility in Slovak local dairy products.

## 2. Materials and Methods

### 2.1. Isolation and Identification of Lactiplantibacillus plantarum (Lactobacillus plantarum) LP17L/1

Stored sheep cheeses were supplied by different dairy plants (34) located in central Slovakia. The standard dilution microbiological method (International Organization for Standardization, ISO) was used to treat samples as follows: sample (10 g) was mixed with 90 mL of peptone water (Merck, Darmstadt, Germany) using a Stomacher–Masticator homogenizer (IUL Instruments, Barcelona, Spain). Then, samples were diluted in Ringer solution (pH 7.0, Merck, Darmstadt, Germany). The dilutions were plated on MRS medium (Merck, Darmstadt, Germany) and cultivated at 37 °C for 48 h. Randomly picked colonies were checked for purity on MRS medium enriched with 5% of sheep blood. Then, pure colonies were submitted for taxonomical identification by the MALDI-TOF identification system (Bruker Daltonics, Billerica, MA, USA) based on the analysis of the bacterial proteins [15]. This was performed using a Microflex MALDI-TOF mass spectrophotometer as also described by Lauková et al. [5]. Lysates of bacterial cells were prepared according to the producer’s recommendation (Bruker Daltonics, Billerica, MA, USA). Results were evaluated using the MALDI Biotyper 3.0 (Bruker Daltonics, Billerica, MA, USA) identification database. Taxonomic allocation of strains was evaluated on the basis of highly probable species identification (score 2.300–3.000), secure probable species identification/probable species identification (2.000–2.299), and probable genus identification (1.700–1.999). The *L. plantarum* DSM 20633T DSM strain involved in the identification system database was a positive control. The identified *Lpb. plantarum* LP17L/1 strain was stored using the Microbank system (Pro-Lab Diagnostic, Richmond, Canada) for further analyses. Genotypization of *Lpb. plantarum* was performed according to an amplification protocol [16] with the following primers: 5′-ATGAGGTATTCAACTTATG-3′ and 5′-GCTGGATCACCTCCTTTC-3′ according to the following program: amplification consisted of 30 cycles 1 min at 94 °C, 1 min at 53 °C, and 1 min at 72 °C. The first cycle was preceded by incubation for 5 min at 94 °C. Visualization was performed by agarose electrophoresis (0.8% agarose, Sigma-Aldrich Chemie GmbH, Schnelldorf, Germany). The control strain was *L. plantarum* CCM 4000 (Dr. Nemcová, University of Veterinary Medicine and Pharmacy in Košice, Slovakia).

Phenotypization was provided using the commercial kit BBL Anaerobe (Becton and Dickinson, Cockeysville, MD, USA), and compared with a reference strain according to de Vos et al. [17] following the fermentation of carbohydrates (arabinose, cellobiose, mannitol, melibiose, raffinose, fructose, ribose, sucrose, xylose), esculin hydrolysis, and others such as L-serine AMC, L-isoleucine-AMC, L-methionine-AMC, p-n-p-phosphate, L-phenylalanine-AMC, or 4MU-β-D-xyloside.

Based on technological properties such as the acidify ability test, coagulate formation during the growth in raw milk, non-production of biogenic amines (2-fenyletylamine, putrescin, cadaverin, histamine, tyramine, spermidine, spermin) as well as non-CO_2_ production, the LP17L/1 strain was selected for additional study [18].

### 2.2. Hemolysis, Antibiotic Phenotype Profile, Biofilm-Forming Ability, and Enzyme Production of Lpb. plantarum LP17L/1

MRS agar (Difco, Sparks, MD, USA) supplemented with 5% of defibrinated sheep blood was inoculated by tested strain to check hemolysis. Plates were incubated at 37 °C for 24 h under semi-anaerobic conditions. The presence of clearing zones around the colonies was interpreted as β, α-hemolysis. The absence of zones was evaluated as a negative result, ᵧ-hemolysis, respectively [19].

The antibiotic phenotype was tested using antibiotic strips [20] with minimal inhibition concentration (MIC in µg) on Mueller-Hinton agar (Merck, Darmstadt, Germany). The control strain was *Lpb. plantarum* CCM 4000 (kindly supplied by Dr. Nemcová, University of Veterinary Medicine and Pharmacy in Košice, Slovakia). Evaluating was performed according to the manufacturer`s instructions. The following strips and concentrations (with established breakpoints) supplied by Oxoid (Oxoid, Basingstoke, UK) were used: ampicillin (0.015–256 µg/mL), penicillin (0.002–32 µg/mL), tetracycline (0.015–256 µg/mL), erythromycin (0.015–256 µg/mL), vancomycin (0.015–256 µg/mL), oxacillin, and gentamicin (0.064–1024 µg/mL).

Biofilm-forming ability in the LP17L/1 strain was analyzed using the quantitative plate assay by Chaieb et al. [21] and Slížová et al. [22]. One colony of the LP17L/1 strain grown on MRS (Merck, Darmstadt, Germany) overnight at 37 °C was transferred into 5 mL of Ringer solution (pH 7.0, 0.75% *w*/*v*) to obtain a suspension corresponding to 1 × 10^8^ CFU/mL. This dilution (100 µL volume) was transferred into 10 mL of MRS broth (Merck, Darmstadt, Germany). The volume (200 µL) of dilution in MRS was inoculated into polystyrene microtiter plate wells (Greiner ELISA 12 Well Strips, 350 µL, flat bottom, Frickenhausen GmbH, Germany). The microtiter plate was incubated for 24 h at 37 °C. The biofilm formed in the microtiter plate wells was washed twice with 200 µL of deionized water and dried at 25 °C for 40 min. The attached bacteria were stained at 25 °C for 30 min with 200 µL of 0.1% (*m*/*v*) crystal violet in deionized water. After the dye solution was aspirated away, the wells were washed twice with 200 µL of deionized water. The plate was dried at 25 °C for 30 min. The dye bound to the adherent biofilm was extracted with 200 µL of 95% ethanol. A 150 µL volume was transferred from each well into a new microplate well for absorbance (A) measurement at 570 nm. An Apollo 11 Absorbance Microplate reader LB 913 (Apollo, Berthold Technologies, Oak Ridge, TN, USA) was used. Each strain and condition were tested in two independent runs with 12 replicates. A sterile MRS was included in each analysis as a negative control. *Streptococcus equi* subsp. *zooepidemicus* CCM 7316 was used as a positive control (kindly provided by Dr. Eva Styková, University of Veterinary Medicine and Pharmacy in Košice, Slovakia). Biofilm-forming ability was classified as highly positive (A_570_ ≥ 1.0), low-grade positive (0.1 ≤ A_570_ <1.0), or negative (A_570_ < 0.1) [21,22]. 

The following enzymes were tested: alkalic phosphatase, esterase, esterase lipase (C8), lipase (C14), leucine arylamidase, valine arylamidase, cystine arylamidase, trypsin, α-chymotrypsin, acidic phosphatase, naphthol-AS-Bi-phosphohydrolase, α-galactosidase, β-galactosidase, β-glucuronidase, α-glucosidase, β-glucosidase, N-acetyl-β-glucosaminidase, α-mannosidase, and α-fucosidase. They are involved in the API-ZYM system kit (BioMérieux, Marcy l`Etoile, France). Enzyme activities were evaluated according to the manufacturer`s instructions (after 4 h of incubation at 37 °C). Color intensity values from 0 to 5 and relevant values in nanomoles (nmoL) were assigned for each reaction according to the color chart supplied with the kit as previously described by Lauková et al. [23].

### 2.3. Tolerance to Oxgall/Bile, Low pH and Growth in Skim Milk (Biofermentor Biosan)

The tolerance of the LP17L/1 strain in a bile environment was tested in MRS broth (Merck, Darmstadt, Germany) enriched with 1% oxgall/bile (Difco, Sparks, MD, USA) according to Gilliland and Walker [24]. An overnight culture of the LP17L/1 strain was inoculated (0.1%) into MRS broth without and with oxgall/bile and incubated at 37 °C for 180 min. Viable cells of the tested strain (in medium with and without oxgall/bile) were counted at time zero (0), at 90 min, and 180 min. Appropriate dilutions (in Ringer solution, Merck, Darmstadt, Germany) were plated on MRS (Merck, Darmstadt, Germany). Surviving cells of LP17L/1 grown on MRS agar at time zero (0), at 90 min, and at 180 min were expressed in CFU/mL.

Tolerance to pH 2.5 was tested in simulated gastric juice (SGJ) containing pepsin and also not containing pepsin (Sigma-Aldrich Chemie GmbH, Schnelldorf, Germany) according to Arboleya et al. [25]. Tubes containing SGJ with and without pepsin were inoculated with a 0.1% overnight culture of the LP17L/1 strain. Surviving cells were counted on MRS agar at time zero (0), and at 90 min and 180 min as formerly mentioned and expressed in CFU/mL.

Skim milk (Difco, Sparks, MD, USA) in a tube was inoculated with 0.1% pre-inoculum (overnight culture of the LPa17L/1 strain—9.0 × 10^9^ CFU/mL) and cultivated at 37 °C in the Biofermentor Biosan RTB-1C (Laboserv Company, Brno, Czech Republic). The growth of the strain (absorbance, A_600_) was measured every 30 min from time 0 (before cultivation) for 24 h. After 24 h, the cell count was calculated by spreading an appropriate dilution on MRS agar (Merck, Darmstadt, Germany) and the cells count was expressed in CFU/mL.

### 2.4. Structural Plantaricin Genes Analysis and Bacteriocin Activity of LP17L/1

The genes for 10 *plantaricins* were analyzed. The plantaricin (*pln)* A, B, C, N, and K genes were amplified through a PCR multiplex adapted from Remiger et al. [26] and Sáenz et al. [27] in a thermocycler VWR Dopio (VRW International, Leuven, Belgium). PCR amplification of other gene sequences involved in plantaricin production was carried out using the primers and conditions specified in Table 1. *Lactobacillus plantarum* ATCC BAA793^TM^ was used as a positive control.

Firstly, the bacteriocin activity of LP17L/1 was tested using the qualitative method according to Skalka et al. [28]. Briefly, an MRS agar plate (Merck, Darmstadt, Germany) inoculated with the tested strain was incubated at 37 °C overnight. After that, the plate was overlaid with 4 mL of soft agar (0.7%) and seeded with 200 µL of indicator bacteria (overnight culture-absorbance measured at 600 nm-A_600_ up to 0.8 which corresponds to 10^6^–10^9^ CFU/mL based on the type/species of indicator strain). The plates were incubated overnight and widths of the clear inhibitory zones were measured in mm. In this testing, the most susceptible indicator -*Enterococcus avium* EA5 (fecal isolate of our laboratory) was used, followed by listeriae; *Listeria monocytogenes* CCM4699-clinical strain, the strains of *L. monocytogenes* P2024, P7223, P7562-isolated from various meat products (State Veterinary and Food Institute in Olomouc, Czech Republic), *L. innocua* LMG13568 (University Brussel, Belgium), *Staphylococcus aureus* SA5 (isolated from mastitis milk), *S. aureus* SABok1, and *S. aureus* SASedl4 (from ewe’s milk lump cheeses). These species were selected as the most frequent contaminants in raw milk. To obtain a more concentrated bacteriocin substance, the supernatant of the LP17L/1 strain was concentrated using Concentrator plus (Eppendorf AG, Hamburg, Germany). Briefly, the LP17L/1 strain (0.1% pre-inoculum) was inoculated in MRS broth (Merck, Darmstadt, Germany) and incubated overnight at 37 °C. Then, the culture was centrifuged for 30 min at 10,000× *g* (at laboratory temperature). The supernatant was concentrated 20 × fold to achieve a volume of 4 mL. The inhibitory activity was tested using the method according to De Vuyst et al. [29] against *E. avium* EA5, 12 strains of *L. monocytogenes* from various food products, *L. monocytogenes* CCM4699 (clinical strain, Czech Culture Collection, Brno, Czech Republic), *L. innocua* LMG13568 (University Brussel, Belgium), and 10 strains of *S. aureus* (from mastitis milk-1, from the feces of rabbits, and intestines of trout). Bacteriocin activity was expressed in arbitrary units per mL, indicating the highest dilution of concentrated bacteriocin substance which can inhibit the growth of the indicator strain.

**Table 1 foods-11-00959-t001:** The primers used for plantaricin genes detection.

Target	PCR Primers	Amplicons (pb)	Temperature of Annealing	References
*pln* A	F-GTACAGTACTAATGGGAG			
	R-CTTACGCCATCTATACG	450	53.5	[26,27]
*pln* B	F-GCTTCTTATTTAAGTAGAGGATTTCTG			
	R-GCCACGATTACTACCCTTAG	927	53.5	
*pln* C	F-AGCAGATGAAATTCGGCAG			
	R-ATAATCCAACGGTGCAATCC	108	53.5	
*pln* D	F-TGAGGACAAACAGACTGGAC			
	R-GCATCGGAAAAATTGCGGATAC	414	54	
*pln* K	F-CTGTAAGCATTGCTAACCAATC			
	R-ACTGCTGACGCTGAAAAG	246	53.5	
*pln* J	F-TAACGACGGATTGCTCTG			
	R-AATCAAGGAATTATCACATTAGTC	475	53.5	
*pln* L	F-ACGGCGTCTGAGATCCAATG			
	R-GTTCTGGAAGTCACTGCGATTG	413	56.5	
*pln* M	F-AAGCGGTATATTAAAAGCGTAGAG			
	R-CATTTCCTCCTTAAAGCATTCAAC	444	54	
*pln* N	F-ATTGCCGGGTTAGGTATCG			
	R-CCTAAACCATGCCATGCAC	46	35.5	
*pln* R	F-CCCAGCAGTCCCATCACTAA			
	R-TTACGGAGCGGCATCTATGTC	236	56.5	

### 2.5. To Confirm Proteinaceous Character of Bacteriocin Substance

MRS broth (Merck, Darmstadt, Germany) was inoculated with *Lpb. plantarum* LP17L/1 and incubated overnight at 37 °C. Then, it was centrifuged at 10,000× *g* for 30 min (at laboratory temperature). The supernatant was taken away and the cells (10^9^ CFU/mL) were dissolved in Ringer solution. The surface of the 1.5% agar plate was overlaid with BHI soft agar (*v*/*w*, 0.7%) containing 200 µL of indicator strain. After drying, 5 µL of LP 17L/1 cells was dropped on the agar plate surface. To each drop was added 2–3 µL of the enzyme protease K (10 mg/mL). Plates were incubated at 37 °C overnight. When a clear halfmoon-formed zone appeared, it indicated the proteinaceous character of the substance. The protease-treated substance was then checked against indicator bacteria such as *S. aureus* from chondritis (R. Nemcová, UVMP in Košice) and trout (our isolates) as well as against *S. aureus* SA5, *S. aureus* SABok1, SASedl4, Kek2, and *L. monocytogens* P 7223. Indicator bacteria used were overnight culture-absorbance measured at 600 nm-A_600_ up to 0.8 which again corresponds to 10^6^–10^9^ CFU/m based on the type/species of the indicator strain.

### 2.6. In Vivo Safety Control and Effect against Coliform Bacteria

To test the in vivo safety of the LP17L/1 strain and its inhibitory effect against coliforms, aged eight weeks, pathogen-free male Balb/c mice (VELAZ Prague, Czech Republic) were used with weight ranging from 18 g to 20 g. Mice were kept under a 12 h light/dark regimen at a temperature of 22–24 °C with a humidity of 56%. They were placed on a commercial diet and water was available without restriction [30]. Animals were divided randomly into 2 groups: control (*n = 15*) and experimental group (*n = 17*). To differentiate the LP17L/1 strain from other LAB, its variant marked by rifampicin was prepared [31]. LP17L/1 was administered per os daily at a total dose of 100 µL (10^9^ CFU/mL). Its count, as well as the count of other LAB, was enumerated after the standard microbiological dilution of feces and jejunum (jejunum was homogenized using the Stomacher–Masticator, Spain) and then plated on MRS agar enriched with rifampicin (100 µg) and MRS agar-rifampicin free. Moreover, coliform bacteria were counted to test the in vivo inhibitory activity due to the LP17L/1 strain. The bacterial count was expressed in CFU/g ± SD. The sampling of feces and jejunum was performed at the start of the experiment (*n = 20*), at day 7 (mixture samples, *n = 5*), and at day 30 (mixture samples *n = 5*.) For the jejunum, mixture sampling was also provided (*n = 3*).

## 3. Results

### 3.1. Identification, Hemolysis, Antibiotic Phenotype Profile, Biofilm-Forming Ability, and Enzyme Production of Lpb. plantarum LP17L/1

Evaluation of the MALDI-TOF system score (2. 387) allotted the identified strain to the species *Lactiplantibacillus plantarum* (previously *Lactobacillus plantarum,* Zheng et al. [6]). A high score was associated with highly probable species identification (2.300–3.000) which was also confirmed by the PCR result and by phenotypization as indicated by Zheng et al. [6]. Genotypization using PCR confirmed the species allocation compared with the *L. plantarum*-positive control strain.

*Lpb. plantarum* LP17L/1 was hemolysis negative (ᵧ-hemolysis) and susceptible to antibiotics (P-2 µg/mL, Amp-ampicillin 25 µg, Tc-tetracycline 12 µg, Gn-gentamicin 16 µg, Ox-oxacillin 4 µg, E-erythromycin, Van-vancomycin 4 µg/mL. *Lpb. plantarum* LP17L/1 does not form a biofilm (0.096 ± 0.002). The values measured for the individual enzyme in the API ZYM test reached zero (no enzyme production) for alkalic phosphatase, esterase, esterase lipase (C8), lipase (C14), leucine arylamidase, valine arylamidase, and cystine arylamidase; no production was also evaluated for the enzyme trypsin, α-chymotrypsin, acidic phosphatase, and naphthol-AS-Bi-phosphohydrolase. A value of 10 nmoL was measured for β-galactosidase, and β-glucosidase. LP17L/1 did not produce β-glucuronidase, N-acetyl-β-glucosaminidase, α-mannosidase, α-fucosidase, and α-galactosidase. In α-glucosidase, 5 nmoL was measured.

### 3.2. Tolerance to Oxgall/Bile, Low pH and Growth in Skim Milk (Biofermentor Biosan), Enzyme Analysis

LP17/1 sufficiently tolerated 1% of oxgall/bile, when after 90 min cultivation, the difference in cell count between the sample with and without oxgall/bile was in the order of 5.6 × 10^6^: 9.0 × 10^7^ CFU/mL and after 180 min cultivation of the cell count of the LP17/1 strain reached 4.0 × 10^5^ in comparison to the control tube (2.0 × 10^6^, Table 2), still one order difference. Initial counting in this analysis was 3.0 × 10^9^ CFU/mL. In the case of the gastric environment, when its count in gastric juice-simulating medium reached 2.7 × 10^4^ CFU/mL after 180 min incubation in comparison with the control (2.0 × 10^8^ CFU/mL), it did not produce damaging enzymes such as trypsin, α- chymotrypsin or β- glucuronidase. In contrast, it produced 10 nmoL of β- galactosidase.

In Biofermentor Biosan, the LP17L/1 strain showed sufficient growth to achieve 7.2 × 10^8^ CFU/mL in skim milk compared to the same time in MRS (9.0 × 10^9^ CFU/mL) for 24 h in MRS.

### 3.3. Structural Plantaricin Genes Analysis, Bacteriocin Activity of LP17L/1 and Proteinaceous Character of Substance

Ten *pln* genes in LP17L/1 strain were confirmed: *pln* A (450 bp), *pln* B (475 bp), *pln* C (108 bp), *pln* D/414 bp), *pln* J, *pln* K, *pln* L, *pln* M, *pln* N (146 bp), and *pln* R.

Using the qualitative method, LP17L/1 showed inhibitory activity against two listeriae, *L. monocytogenes* CCM4699 and *L. innocua* LMG13568 (inhibitory zone 6 mm), out of 8 different indicators such as *E. avium* EA5, *S. aureus* SA5, SABok1, SASedl4, *L. monocytogenes* CCM4699, *L. monocytogenes* P2024, *L. monocytogenes, L. monocytogenes* P7223, and *L. monocytogenes* P7562. Therefore, the substance from the LP17L/1 strain was concentrated using Concentrator plus (Spain) to obtain a more concentrated bacteriocin substance. The inhibitory activity was checked using the quantitative method against 31 indicators (Table 3) [29]. The concentrated substance showed inhibition against 9 out of 14 *L. monocytogenes* strains (Table 3) with inhibitory activity ranging from 100 to 3 200 AU/mL. The growth of *L. monocytogenes* CC4699, *L. innocua* LMG13568, *L. monocytogenes* P7562, P6301, P2116, P2024 was not inhibited using concentrated substance LP17L/1 as well as the growth of *S. aureus* SA5, K1/2, and *E. avium* EA5. However, the growth of 14 out of 16 *S. aureus* strains was inhibited (Table 3). The most susceptible was *L. monocytogenes* P7223. This means that 23 out of 31 indicators (74.2%) were inhibited using the concentrated substance.

After treatment with protease K, half-moon inhibitory zones were evaluated with 15 indicator strains confirming the proteinaceous character of the substance and eliminating inhibition due to other effects (e.g., acid production or other organic substance). The growth of these indicators was inhibited: *L. monocytogenes* P7223, *S. aureus* SA5 (mastitis milk), SABok1, SASedl4 (cheese), SAKek/2, SA31/9, 31/8, 31/4, 31/3, 31/6 (feces of rabbits), and *S. aureus* SA5/3, SA3/1, SA6/1, SA3/4, SA 2/1 (trouts). 

### 3.4. In Vivo Control of Safety and Activity of LP17L/1

The LP17L/1 strain sufficiently colonized mice intestines. At day 7, its count in the feces reached 3.63 ± 1.83 CFU/g log 10 (Table 4). Counts of LAB were high (6.48 ± 2.54 CFU/g log 10) and well balanced compared to day 0/1 (Table 4). At day 30, counts of the LP17L/1 strain decreased; however, the total LAB count was increased (7.38 ± 2.71 CFU/g). The decrease in the LP17L/1 strain can be explained by competitive interaction with other LAB flora. In the jejunum, its sufficient count was also noted (2.07 ± 1.43 CFU/g log 10) and similarly as in the feces at day 30, also in the jejunum, a decrease in the LP17L/1 strain was noted. The total LAB in the jejunum was still sufficient and similarly as in the feces at day 30; they were increased also in the jejunum, while the LP17L/1 count decreased at the same time. However, no mortality or symptoms were noted in mice. Coliforms control in the feces showed no influence, but in the jejunum at day 7, a difference of 2.71 log cycle was found compared to sampling at day 0/1 and it was almost the same at day 30.

## 4. Discussion

The species *Lactiplantibacillus plantarum* is a bacterium from the genus *Lactiplantibacillus*, family Lactobacillacae, order Lactobacillales, class Bacilli, and phylum Firmicutes. Because of the progressive genomic sequencing method, the lactobacilli description of 23 novel genera was evaluated by Zheng et al. [6]. The accession number for type strain *Lpb. plantarum* is AZFR00000000. The identity of our strain was also confirmed by next-generation sequencing showing similarity with the same species strains in the general database up to 91.4% (data not shown, personal communication, Dr. Marián Maďar, UVMP, Košice). The species *Lactiplantibacillus plantarum* can be frequently detected in dairy products. This is also the case with our strain. It is a lactic acid-producing (data not shown) strain, but also a bacteriocin-producing strain. Its sufficient growth in skim milk and tolerance to oxgall/bile and low pH was confirmed in this study. If the strain is tested for its further probiotic use in products for human consumption, it is required to work/function under gastrointestinal (GIT) conditions; this is because the strain can grow sufficiently under those GIT conditions and can better function there [32]. Listeriae are frequent contaminants of dairy products. Because listeriae can also grow at refrigeration temperatures, they can survive easily in the product [33]. Staphylococci can be also often detected in milk and dairy products [33]. One approach to reducing the prevalence of those contaminants is the use of bacteriocins with a broad antimicrobial spectrum [34]. The in situ effects of enterocins (bacteriocins mostly produced by representatives of the genus *Enterococcus*) have been already published, e.g., in skimmed milk or yogurt [34]. A decrease of 8 log cycles in viable cells of *S. aureus* SA1 in skimmed milk was reported by Lauková et al. [35] during 24 h. Using enterocin CCM 4231 was also demonstrated in yogurt (a decrease of 3 log cycles CFU/g). Although in this study non-purified plantaricin was mentioned, the strain *Lpb. plantarum* 17L/1 possessed 10 *pln* genes (plantaricins). Plantaricin MG produced by *L. plantarum* and isolated from Jiaoke, a traditional fermented cream from China, was active against *L. monocytogenes*, *S. aureus, E. coli*, and *S. Typhimurium* [36]. Cukrowska et al. [37] also described lactobacilli with antagonistic activity against listeriae, *S. aureus* ATCC25923 and salmonellae which sufficiently tolerate low pH and oxgall/bile. A mixture of those strains applied in Balb/c mice affected the cytokine TH1/TH2 balance toward a non-allergic TH1 response. Moreover, the best protective effect against *Trichinella spiralis* infection associated with the increased oxidative metabolism of peritoneal macrophages was exhibited by the LP17L/1 strain which activated the metabolic activity of macrophages during the migration of newborn larvae (from day 5 to 25 post infection) [30]. This means the model experiment using Balb/c mice showed that the beneficial strain has the ability to decrease the intensity of parasitic infection by affecting important components of the innate immune system such as phagocytosis [38]. The LP17L/1 strain showed impact as an effective mediator to regulate macrophages’ oxidative metabolism in *T. spiralis* infection, which is promising in trichinellosis treatment or prevention. After an in vitro fecundity test, it was also observed that the LP17L/1 strain revealed a direct inhibitory impact on female fecundity (about 80%) [39]. In this study, the LP17L/1 strain did not cause mortality when applied in Balb/c mice. In addition, in the jejunum, coliform bacteria were decreased. The antimicrobial effect of the beneficial strain can be also increased due to the co-interaction of bacteriocin and lactic acid. *Lpb. plantarum* L7L/1 also produces β- galactosidase which is the enzyme used in the dairy industry for the production of lactose-free milk intended for lactose-intolerant consumers [40]. *Lpb. plantarum* 17L/1 was involved in the utility model (PÚV 50094-2021) which is processed at the Industrial Property Office of the Slovak Republic and deposited in the Czech Culture Collection in Brno, Czech Republic (CCM 9208).

## 5. Conclusions

*Lactiplantibacillus plantarum* LP17L/1 is a non-hemolytic and non-biofilm-forming strain, susceptible to commercial antibiotics, which contains genes for 10 *plantaricins* and exerted active antimicrobial substances of proteinaceous character—bacteriocin showing in vitro anti-staphylococcal and anti-listerial effects. It does not produce damaging enzymes; however, it produces β- galactosidase. It sufficiently survives in the mouse gastrointestinal tract without side effects which confirms its safety. Moreover, its application in Balb/c mice for one month reduced the count of coliform bacteria in the jejunum. Additional tests are in progress; however, LP17L/1 is a promising additive for Slovak local dairy products.

## Figures and Tables

**Table 2 foods-11-00959-t002:** Survival of *Lactiplantibacillus plantarum* LP17/1 in oxgall/bile as well as at low pH expressed in colony-forming units per milliliter (CFU/mL).

	Time of Cultivation	
	90 min	180 min
LP17/1Ox	5.6 × 10^6^	4.0 × 10^5^
Control	9.0 × 10^7^	2.0 × 10^6^
LP17/1pH	4.5 ×10^5^	2.7 × 10^4^
Control	3.8 × 10^8^	2.0 × 10^8^

Initial count of LP17L/1 strain at start 0 h of analysis to survive in oxgall/bile was 3.0 × 10^9^ CFU/mL and in case of pH (2.5) study it was 9.2 × 10^8^ CFU/mL.

**Table 3 foods-11-00959-t003:** Inhibitory activity of concentrated substance LP17L/1 expressed in arbitrary units per mL AU/mL.

Indicators	Inhibitory Activity
*Listeria monocytogenes*	
P3300	100
P5258	200
P6501	100
P7223	3 200
P7395	100
P7401	100
Ve40	100
P10811	100
P7395	100
*S. aureus*	
Kek2	800
Nip/1	100
Rum/1	100
Bel/1	100
31/5	400
31/6	400
33/4	400
39/9	100
39/10	400
SA5/3ch	100
Sa3/1ch	200
SA6/1ch	800
SA3/4ch	200
SA2/1	100

*Listeria monocytogenes* P3300-97395, clinical strains, *L. monocytogenes* CCM4699 (CCM, Brno, CZ), *L. innocua* LMG13568 (Belgium) were not inhibited. *E. avium* EA5 not inhibited, SA5-mastitis milk not inhibited, *Staphylococcus aureus* SA31/5-39/10-trouts, Kek/2-Bel/1- feces of rabbits. The rest of the strains were not inhibited.

**Table 4 foods-11-00959-t004:** In vivo safety and activity of LP17L/1 strain in model experiment with Balb/c mice. Counts are expressed in CFU/g (log 10).

Feces	LP17L/1	LAB	Coliforms
Sampling I *n* = 10 day 0/1	nt	6.96 ± 2.60	3.24 ± 1.84
Sampling II *n* = 5 day 7	3.63 ± 1.83	6.48 ± 2.54	3.60 ± 1.89
Sampling III *n* = 5 day 30	1.21 ± 0.10	7.38 ± 2.71	3. 60 ± 1.89
Jejunum Sampling I *n* = 3 day 0/1	nt	4.98 ± 2.20	4.21 ± 2.05
Sampling I *n* = 5 day 7	2.07 ± 1.43	4.19 ± 2.04	1.50 ± 0.22
Sampling I *n* = 5 day 30	<1.0	5.23 ± 2.28	1.15 ± 0.07

NS, only mathematical difference, not significant/statistical difference; coliforms, day 0/1 to day 7 and 30; LAB, lactic acid bacteria; nt-not tested;

## Data Availability

Not applicable.

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
