# Peer review of "Bacteriocin-Producing Strain Lactiplantibacillus plantarum LP17L/1 Isolated from Traditional Stored Ewe’s Milk Cheese and Its Beneficial Potential"

_foods, 2022, doi:10.3390/foods11070959_

Round 1

Reviewer 1 Report

General

The work is devoted to the study of a strain of Lactiplantibacillus plantarum LP17L/1 isolated from traditional stored 3 ewe`s milk cheese which produce antimicrobial substances, including bacteriocin. Along with other substances, bacteriocin has a wide range of antibacterial properties against pathogenic foodborne bacteria, as well as to extend the shelf life of foods.

The subject is interesting, but I think that the manuscript needs some corrections.

Specific comments:

  1. What are the subsequent or intended ways of using the strain of Lactiplantibacillus plantarum LP17L/1 for food industry?

Page 6, Line 5: Please clarify the number of indicator bacteria cells

Page 6, Line 37: Please clarify the number of indicator bacteria cells  

  1. Page 7, Lines 22-28: Explain more clearly to which antibiotics the strain was sensitive, and what the MIC concentrations were for Lactiplantibacillus plantarum LP17L/1. The authors indicate a very wide range of MICs for one strain, which raises doubts about the results.

  1. Page 7, Lines 31-37: Please clarify what is the difference between zero enzyme level and no enzyme. Specify which enzymes were ultimately active in Lactiplantibacillus plantarum LP17L/1

  1. Page 9. Table 3: Can't find results in table «Inhibitory activity of concentrated substance after its treatment with protease K (b)».

  1. Page 9. Line 31: English correction “wer enot inhibited”
  2. Page 10. Table 4: Please express the numbers in the table as CFU/g; please clarify - «only mathematical difference»
  3. Page 11. Line 15: English correction “plnataricin”

Author Response

Responses to reviewers Foods-1640678

First of all I thank so much to reviewers giving me chance to improve the manuscript.

Reviewer 1 (in red)

  1. Because plantarum can be used, that one strain with probiotic or bacteriocinogenic potential as/for starter culture as funcional additive for functional product. Because taxonomy of lactobacilli was upgraded in 2021, and it is also regarding L. plantarum species, the strain LP17/1 is now validated as Lactiplantibacillus plantarum, so it has promising potential as dairy industry additive culture.

P6, line 5 clarify the number of indicator bacteria cells....... of indicator bacteria  (overnight culture-absorbance measured at 600 nm-A600 up 0.8). It means A=0.8 corresponds  with  106-109 CFU/mL based on the type/species of indicator strain.

2.P7, lines 22-28, It was missunderstanding, those MIC ranges are showed for commercial strips. Results regarding the strain susceptibility are  as follows: susceptible to antibiotics (P-2 µg/mL, Amp-ampicillin 25 µg, Tc-tetracycline 12 µg, Gn-gentamicin 16 µg, Ox-oxacillin 4 µg, E-erythromycin, Van-vancomycin 4 µg/mL which can show to followers  susceptibility or resistance of strain. It is MIC ranges for those antibiotics for lactobacilli. So, from results it is clearly seen.

3.P7, lines 31-37, zero and no enzyme it is the same meaning no enzyme was produced.

  1. p9/Table 3 I apologize, here were involved antimicrobial inhibitory activity of concentrated substance and treated with protease are not involved in the Table 3, because not all indicators or others were used in the test with bacteriocin substance treated with protease, so it is involved only in the results text.

5.p9, 31, 6.p10, Table 4, 7.p11, line 15-corrected/revised

Reviewer 2 Report

Dear Editors and authors.

Although the production of bacteriocins from lactic acid bacteria has been a fascinating topic in recent years, this manuscript contains numerous scientific errors.

1-The abstract of the manuscript contains a long introduction that is not needed. It  should contain the best results and the most important conclusions.

2-he manuscript mentions the creation of bacteriocins from Lactiplantibacillus plantarum bacteria, but there is no mention of bacteriocins, their manufacture, or their usage in the introduction.
I recommend reading the paper (Verma, D. K., Thakur, M., Singh, S., Tripathy, S., Gupta, A. K., Baranwal, D., ... & Srivastav, P. P. (2022). Bacteriocins as antimicrobial and preservative agents in food: Biosynthesis, separation and application. Food Bioscience, 101594.‏).

3-Page 2 lin2 94-95, Where the program of PCR?

4-Page 3 line102, What is the other test?

5-Page 3 line 111, How was the colour distinguished? The colour of the cultural media is brown.

6-Page 3 line 116, This method is a big error. Why did not they use the Mueller-Hinton agar?

7-Many working methods do not cite the references like tolerance to Oxgall/Bile, Low pH and Growth in skim milk.

8-I mentioned many methods of working, but there are no results in the results chapter like Structural Plantaricin Genes Analysis and Genotypization of Lpb. plantarum

Author Response

Responses to reviewers Foods-1640678

First of all I thank so much to reviewers giving me chance to improve the manuscript.

Rev. 2

Thanks for advices, Abstract was re-written.

Introduction (it was revised as recommended.

Bacteriocins are low molecular, thermos-stable, antimicrobial and ribosomal active peptides which are synthesized by many species of bacteria including lactic acid bacteria (LAB). They show antimicrobial activities against food pathogens [10, 11]. Many lactobacilli with probiotic character are described as functional additives for dairy products [12, 13]. Molecular studies reveal that bacteriocin determinants are mostly grouped in operons-regulons [11]. Structural genes for bacteriocin production can be found on plasmids, on a transposon or on mobile genetic element inside the bacterial chromosome [14]. Bacteriocins produced by different food-originated LAB of many species have been reported up to now.   Their benefit is not only as additive to beneficial influence the product itself but they can be beneficial for consumers via product as e.g. cholesterol reducing functional bacteria [12]. Bacteriocin-producing bacteria are used as the main starter or adjunct culture for making cheese or fermented dairy products to prevent spoilage bacteria [11]. In dairy industry were mostly used those strains of the species Lactococcus lactis, and/or Lactobacillus casei [11]. This study was focused on antimicrobial potential of Lpb. plantarum strain LP17L/1 originated from stored ewe`s milk lump cheese with its further application possibility in Slovak local dairy products.

P2, lines 94-95 program for PCR: Genotypization of Lpb. plantarum was performed according to amplification protocol [13] with the following primers: 5`-ATGAGGTATTCAACTTATG-3` and 5`-GCTGGATCACCTCCTTTC-3` according to the following program: amplification consisted of 30 cycles 1 min at 94 ℃, 1 min  at 53 ℃, and 1 min at 72 ℃. The first cycle was preceded by incubation for 5 min at 94 ℃.  Visualized was done by agarose electrophoresis (0.8% agarose, Sigma-Aldrich, Germany). The control strain was L. plantarum CCM 4000 (Dr. Nemcová, University of Veterinary Medicine and Pharmacy in Košice, Slovakia).

P3, line 102….. comparing with reference strain according to de Vos et al. [14] following fermentation of carbohydrates (arabinose, cellobiose, mannitol, melibiose, raffinose, fructose, ribose, sucrose, xylose), esculin hydrolysis and others such as  L-serine AMC, L-isoleucine-AMC, L-methionine-AMC, p-n-p-phosphate, L-phenylalanine-AMC or 4MU-beta-D-xyloside.

P3, line 111 How colour was distinguished, Colour of media is brown. I don’t know surely what do you mean. Blood agar is not brown, it is dark red and clearing zones are easy to see, if hemolysis is present.

P3, line 116….I don’t think that it is big error because not all strains can sufficiently grow on Mueller-Hinton agar which is of course, base medium checking antibiotic susceptibility or resistance and recommended also by CLSI and EFSA. We used both media at this analysis. Because growth was better on MRS, so it was involved in the manuscript, but I can also mentioned only Mueller-Hinton agar. But it is necessary to modify pH of medium for lactobacilli.

  1. Working methods such as tolerance to oxgall/bile, low pH or growth skim milk miss references. Maybe you did not noticed, but for oxgall/bile and also for low pH it was involved in original version: Tolerance of LP17L/1 strain in bile environment was tested in MRS broth (Merck, Darmstadt, Germany) enriched with 1% oxgall/bile (Difco, Sparks, Maryland, USA) according to Gilliland and Walker [20].

Tolerance to pH 2.5 was tested in simulated gastric juice (SGJ) containg pepsin and also not containing pepsin (Sigma-Aldrich) according to Arboleya et al. [21].

Regarding growth in skim milk-it was not described previously, in our previous articles, growth of skim milk was analyzed by taraditional standard cultivation, so I cannot there use reference.

  1. Results genotypization and structural genes: It was more clearly involved there: Genotypization using PCR confirmed the species allocation comparing with L. plantarum positive control strain.

Ten pln genes in LP17L/1 strain were confirmed: pln A (450bp), pln B (475bp), pln C (108 bp), pln D /414 bp), pln J, pln K, pln L, pln M, pln N (146 bp), and pl nR.

 We decided not to use Elfo pictures/figure as documentation because I think they not represent high  standard quality for publication but they showed all results done using PCR.

Round 2

Reviewer 2 Report

Dear Editor, 

Authors did all necessary changes to improve the manuscript and now I recommend it for publication in the current form.